# Bioinformatics and Screening of a Circular RNA-microRNA-mRNA Regulatory Network Induced by Coxsackievirus Group B5 in Human Rhabdomyosarcoma Cells

**DOI:** 10.3390/ijms23094628

**Published:** 2022-04-22

**Authors:** Jing Li, Peiying Teng, Fan Yang, Xia Ou, Jihong Zhang, Wei Chen

**Affiliations:** Medical School, Kunming University of Science and Technology, Kunming 650500, China; jinglili0105@163.com (J.L.); tengpeiying7@163.com (P.T.); yangfan@kust.edu.cn (F.Y.); ouxia@kust.edu.cn (X.O.)

**Keywords:** Hand, foot and mouth disease (HFMD), Coxsackievirus Group B5 (CVB5), circRNA-miRNA-mRNA network, viral replication, innate immune response

## Abstract

Hand, foot and mouth disease (HFMD) caused by Coxsackievirus Group B5 (CVB5) is one of the most common herpetic diseases in human infants and children. The pathogenesis of CVB5 remains unknown. Circular RNAs (CircRNAs), as novel noncoding RNAs, have been shown to play a key role in many pathogenic processes in different species; however, their functions during the process of CVB5 infection remain unclear. In the present study, we investigated the expression profiles of circRNAs using RNA sequencing technology in CVB5-infected and mock-infected human rhabdomyosarcoma cells (CVB5 virus that had been isolated from clinical specimens). In addition, several differentially expressed circRNAs were validated by RT-qPCR. Moreover, the innate immune responses related to circRNA-miRNA-mRNA interaction networks were constructed and verified. A total of 5461 circRNAs were identified at different genomic locations in CVB5 infections and controls, of which 235 were differentially expressed. Gene Ontology (GO) and Kyoto Encyclopedia of Genes and Genomes (KEGG) enrichment analysis demonstrated that the differentially expressed circRNAs were principally involved in specific signaling pathways related to ErbB, TNF, and innate immunity. We further predicted that novel_circ_0002006 might act as a molecular sponge for miR-152-3p through the IFN-I pathway to inhibit CVB5 replication, and that novel_circ_0001066 might act as a molecular sponge for miR-29b-3p via the NF-κB pathway and for the inhibition of CVB5 replication. These findings will help to elucidate the biological functions of circRNAs in the progression of CVB5-related HFMD and identify prospective biomarkers and therapeutic targets for this disease.

## 1. Introduction

Coxsackievirus Group B5 (CVB5) is a non-enveloped virus with linear single-stranded RNA that belongs to a member of the genus Enterovirus from the Picornaviridae family. The CVB5 genome is approximately 7400 bp long, with only one open reading frame (ORF), which ultimately constitutes of four structural proteins (VP1, VP2, VP3, and VP4) and seven non-structural proteins (2A, 2B, 2C, 3A, 3B, 3C, and 3D) [1]. CVB5 is one of the common pathogens causes of hand, foot and mouth disease (HFMD) [2,3]. HFMD is generally considered to be a common global exanthemata and febrile illness that is mostly seen in children under 5 years old. This disease is characterized by a maculopapular rash or blisters on the hands, feet, groin, and buttocks and is associated with painful ulcerative lesions of the mouth [4]. Moreover, CVB5 is a neurotropic virus, tends to be more severe, and is more likely to produce severe complications in the central nervous system (CNS), thus causing infections such as aseptic meningitis or, acute brainstem encephalitis. Children who recover from brainstem encephalitis can be left with significant neurological after-effects [5,6]. Thus, exploring the mechanisms of the CVB5 infection might provide new insights into the pathology of HFMD and novel targets for the treatment of meningitis.

Circular RNAs (circRNAs) represent an emerging type of RNA with single-stranded RNA and is expressed in almost all species. Unlike linear RNAs, circRNAs are characterized by back-splicing and the terminated 5′ caps and 3′ tails are not found within circRNAs [7]. MicroRNAs (miRNAs) are sponged by circRNAs as competitive endogenous RNAs (ceRNAs), thus preventing miRNAs from binding to their target mRNAs and resulting in the increased expression of targets, such as circRNAs, which could suppress the production of viral proteins and promote viral clearance by binding miR-122 during a hepatitis C virus (HCV) infection [8]. This regulatory mechanism is the most common mechanism for circRNAs. In addition, RNA-sequencing is revealing an increasing number of circRNA expression profiles after virus infection; evidence suggests that some circRNAs can regulate host–virus interaction through an innate immune response. For example, Qu et al. found that circRNA-AIVR mediated the expression of the CREB Binding Protein (CREBBP) by binding to miRNA, which promoted the production of IFN-β and antagonized the replication of the influenza virus [9]. CircRNA regulated immune responses in viral infections have gradually been demonstrated, but without a focus on the functions and mechanisms involved in the regulation of pathogenesis for CVB5 infection.

In this study, we performed a circRNA expression analysis of CVB5-infected human rhabdomyosarcoma (RD) cells. Gene Ontology (GO) and Kyoto Encyclopedia of Genes and Genomes (KEGG) analyses were adopted to predict potential physiological activities and related signal pathways. We also performed a circRNA-miRNA-mRNA interaction analysis. To better understand host-CVB5 interactions, we determined that novel_circ_0002006 and novel_circ_0001066 exerted effects on viral replication, and we investigated their effects on target miRNA expression; thus, we report a comprehensive catalog of circRNAs and provide a transcriptome to identify novel molecular targets and pathways for the treatment of HFMD.

## 2. Results

### 2.1. Identification of circRNAs Expression Profiles

To identify the circRNA expression profiles associated with CVB5 infection, we obtained sequencing data through the Illumina sequencing platform. As shown in Appendix A, a total of 5461 circRNAs were identified in all samples, and information relating to sequence characteristics was obtained. Of these, 3331 circRNAs were observed to be commonly expressed in both treatment groups; however, 493 and 1637 circRNAs were identified as being exclusively expressed in CVB5-infected cells and mock-infected cells, respectively (Figure 1A). The identified circRNAs were evenly distributed across different chromosomes, except for chromosome Y (Figure 1B). The circRNAs were mainly derived from exons, with a small proportion formed by introns and intergenic regions (Figure 1C). The circRNA sequences ranged from 26 nucleotides (nt) to 1323 nt in length, and the majority of circRNAs were 100 nt to 500 nt in length (Figure 1D).

### 2.2. Functional Analysis for Differential Expression of circRNAs

The circRNA expression levels in CVB5-infected and mock-infected RD cells were compared. There were 235 differentially expressed circRNAs (63 were upregulated and 172 were downregulated) for which the expression level changed more than twofold after CVB5 infection, relative to mock-infected cells (Figure 2A). Inter-sample correlation analysis demonstrated that there was an obvious distinction in circRNA expression patterns between the two groups. Only two circRNAs were simultaneously detected in the cells for different treatments. Some changes to circRNA expression involved a change from absent to present (Figure 2B).

To reveal the biological functions of the differentially expressed circRNAs, GO assignments and KEGG pathway analysis were used to analyze the functions of the host genes of differentially expressed circRNAs. The most significantly enriched GO terms for a cellular component (CC) were ‘intracellular’ and ‘intracellular part’. The biological process categories were mainly involved in the ‘cellular macromolecule metabolic process’ and ‘macromolecule metabolic process’ (Figure 2C, Appendix A). KEGG pathway analysis indicated that differentially expressed circRNAs were mainly enriched in ubiquitin mediated proteolysis, tight junctions, the ErbB signaling pathway, and the TNF signaling pathway (Figure 2D, Appendix A).

### 2.3. Validation of Changes in the Differential Expression of circRNAs after CVB5 Infection

To validate the differential expression of the RNA-seq results, we randomly selected six circRNAs (three upregulated and three downregulated circRNAs) for qPCR validation according to the expression heat map (Figure 3A, Appendix A). As shown in Figure 3B, these results indicated that the novel_circ_0010492, hsa_circ_0075346, novel_circ_0001066 were upregulated, whereas hsa_circ_0008378, hsa_circ_0004390, and novel_circ_0002006 were downregulated. All results were consistent with the results arising from RNA-seq (Figure 3C).

### 2.4. Analysis of a circRNA-miRNA-mRNA Interaction Network

To investigate the potential for circRNAs to act as ceRNAs in a CVB5 infection, we selected five enriched circRNAs (hsa_circ_0008378, hsa_circ_0031831, novel_circ_0002006, hsa_circ_0075346, and novel_circ_0001066) that were involved in the innate immune pathway based on GO and KEGG analyses and established two circRNA-miRNA-mRNA interaction networks. Four circRNAs (hsa_circ_0008378, hsa_circ_0031831, novel_circ_0002006, hsa_circ_0075346) in the IFN signaling pathway were found to target 42 miRNAs and 129 target genes (Figure 4A, Appendix A). Meanwhile, novel_circ_0001066 was found to be involved in the NF-κB signaling pathway-related network and was associated with 19 miRNAs and 78 target genes (Figure 4B, Appendix A). These data show that the circRNA-miRNA-mRNA axes may form a complex interaction network to play a role in CVB5 infection.

### 2.5. Potential Treatment Values of novel_circ_0002006 and novel_circ_0001066 in CVB5 Infection

We transfected pcDNA3.1_novel_circ_0002006 and pcDNA3.1_novel_circ_0001066 into RD cells, respectively. The results showed that novel_circ_0002006 significantly inhibited CVB5 replication, and upregulated PPRs (RIG-I and MDA5), IFN-β, and some critical interferon-stimulating factors (ISGs) (OASL, MXA, ISG15, IFIT1, IFIT2, and IFITM3) (Figure 5A,B). Moreover, novel_circ_0002006 also significantly inhibited CVB5 replication and upregulated inflammatory factors (TNF-α and IL-6) (Figure 5C,D). These results suggest that the two circRNAs exhibit antiviral effects through different innate immune pathways.

### 2.6. Verification of miRNA from Immune Interaction Network

Using RT-qPCR to detect target miRNAs, according to novel_circ_0002006 and novel_circ_0001066 networks, found that the expression levels of miR-4526 and miR-3123 were upregulated while those of miR-152-3p were inhibited in terms of novel_circ_0002006 overexpression (Figure 6A). The overexpression of novel_circ_0001066 inhibited miR-29b-3p and miR-567 but promoted miR-15b-3p (Figure 6B). These results indicated that complicated circRNA-miRNA-mRNA networks were active during CVB5 infection and suggested that these interactive networks regulated viral replication.

## 3. Discussion

CircRNAs have received widespread attention in the field of RNA due to their highly stable closed-loop structure and highly conserved nature [10]. HFMD caused by viral infection is one of the most common diseases in human infants and children [4]. Remarkably, circRNAs are closely associated with viral infections [11,12,13]. CircSIAE targets TAOK2 via miR-331-3p sponge adsorption and inhibits Coxsackie virus B3 proliferation [14]. In our current study, we systematically analyzed the expression profiles of circRNAs infected with CVB5 in RD cells. In addition, we constructed circRNA-miRNA-mRNA networks focused on innate immune response. Moreover, the effect of novel_circ_0002006 and novel_circ_0001066 on CVB5 replication were verified, and the expression of target proteins downstream of the enrichment pathway (IFN and NF-κB signaling pathway) were tested. Furthermore, we verified the possibility that circRNA acted through sponges.

We used RNA-seq to reveal the expression profiles of circRNAs during CVB5 infection. 5461 circRNAs were identified from different genomic locations. Our results further suggested that most circRNAs originate from exons (Appendix A), a finding that is consistent with the spectrum of circRNAs that have been found to be associated with CVA16 and EV71 infection [15,16]; therefore, we speculated that this may be related to the virus rather than cell specificity. Furthermore, the length distribution of circRNAs was consistent for the majority of circRNA expression profiles, except for some circRNAs with more than 1000 nucleotides, although this difference may depend on the specificity of the virus or host tissue. Compared with controls, we identified 235 circRNAs that were differentially expressed after CVB5 infection (Appendix A). KEGG analysis of differentially expressed circRNA host genes identified 100 pathways, for which the *p*-value was < 0.05. We found that host genes corresponding to differentially expressed circRNAs were mainly involved in ubiquitin mediated proteolysis, the ErbB signaling pathway, aminoacyl-tRNA biosynthesis, and the TNF signaling pathway. These results suggested that circRNAs may be involved in the regulation of these cellular processes. This was unlike the circRNA enrichment pathway associated with EV71 and CVA16 infections in SH-SY5Y cells, such as the Wnt signaling pathway, angiogenesis, the p53 pathway, and the PDGF signaling pathway [15,16]. This may be related to the cellular specificity of the host’s circRNA. RD cells are often used to isolate the HFMD virus from clinical specimens, and are also highly susceptible to CVB5 infection. Since the expression of circRNAs underwent alterations after CVB5 infection, it appeared that circRNAs were associated with CVB5 specific replication. Furthermore, the enriched pathways were also demonstrated to be closely associated with the viral infection process. For instance, ubiquitin-mediated protein hydrolysis has been shown to be involved in a number of viral processes, including viral replication, maturation, and release [17,18]. Our current findings may provide new avenues to explore the pathogenesis of the CVB5 infection.

CircRNAs can act as “miRNA sponges”, thereby inhibiting the binding of miRNAs to target genes and thus regulating the expression of mRNAs in certain physiological and pathogenic activities; therefore, the elucidation of circRNA-miRNA-mRNA networks after CVB5 infection in host cells is important for understanding the pathogenesis of CVB5. The host innate immune system is the first line of defense against viral invasion; however, viruses can successfully survive via immune escape. The interaction of the IFN signaling pathway with viral infections has been extensively documented, for example, in the Hepatitis B virus (HBV), Influenza A viruses (IAV), and the Japanese encephalitis virus (JEV) [19,20,21]. Thus, we focused on circRNAs that were enriched in innate immune system processes, and constructed circRNA-associated circRNA-miRNA-mRNA networks by probing potential miRNA binding sites. Hsa_circ_0008378, hsa_circ_0031831, novel_circ_0002006, and hsa_circ_0075346 were involved in the IFN signaling pathway-related network in which 129 mRNAs competed for 42 miRNAs. The novel_circ_0001066 was involved in the NF-κB signaling pathway-related network, in which 78 mRNAs competed for 19 miRNAs. Of these, we focused on miR-152-3p, which plays a role in multiple biological processes. It has been shown that the downregulation of miR-152 can induce impaired hepatic gluconeogenesis by targeting PTEN, which is involved in miR-152-3p-mediated hepatocyte gluconeogenesis via the regulation of the AKT/GSK pathway [22]. In addition, miR-152-3p was aberrantly expressed in hepatitis B virus-associated hepatocellular carcinoma [23]. Furthermore, miR-152-3p plays an important role in regulating the cell cycle, controlling tumor growth, and reducing apoptosis [23,24,25].

In addition, there is accumulating evidence that inflammation is a major driving force in virus infection. For example, circ_ 0000220 and miR-326-3p have been shown to regulate the production of inflammatory cytokines during infection [26]. In the circRNA-miRNA-mRNA triple network diagram, novel_circ_0001066 was involved in the NF-κB signaling pathway-related network in which 78 mRNAs competed for 19 miRNAs. With regard to miR-29b-3p, IFN-I signaling-related miRNAs (ISR-miRNAs), have improved significantly during acute severe acute respiratory syndrome coronavirus-2 (SARS-CoV-2) infection [27]; however, the role of miR-29b-3p in inflammation is yet to be investigated; therefore, the altered expression profiles of circRNAs in CVB5-infected cells may exert their biological functionality by interacting with specific microRNAs, for instance, miR-152-3p with miR-29b-3p; however, the expression of miR-152-3p and miR-29b-3p can also affect viral replication. There are currently neither characteristic nor functionality reports about novel_circ_0002006 and novel_circ_0001066.

By performing an in-depth study of circRNA function, it is possible to gain a better understanding of the mechanism of disease occurrence and improve the prevention and treatment of viral diseases. Based on these complex networks, we validated the role of circRNAs on CVB5 infection and the expression of related pathway target genes. We found that novel_circ_0002006 dramatically inhibited the expression of CVB5 VP1 via the activation of ISGs by using the IFN signaling pathway. Novel_circ_0001066 acted via the NF-κB signaling pathway to influence the inhibitory effect of inflammatory factors on CVB5 infection. These results suggested that novel_circ_0002006 and novel_circ_0001066 are representative of potential inhibitors of CVB5 infection. Furthermore, circRNAs have regulatory effects on NF-κB-related and IFN-related IRF3 with ISGs.

In summary, our data demonstrated expression profiles of circRNAs during CVB5 infection RD. Differentially expressed circRNAs may be involved in the mechanisms underlying innate immunity and could act as microRNA sponges to regulate the expression of mRNAs. These results may facilitate future studies investigating the molecular functions of circRNAs in viral pathogenesis.

## 4. Materials and Methods

### 4.1. Cells, Viruses, and Infection

Human striated muscle sarcoma cells (RD, American Type Culture Collection, CCL-136), the green monkey kidney cell line (Vero, American Type Culture Collection, CCL-81), and human embryonic kidney 293T (HEK-293T, American Type Culture Collection, CRL-3216) cells, were cultured in Dulbecco’s modified Eagle medium (DMEM, HyClone, Logan, UT, USA), supplemented with 10% heat-inactivated fetal bovine serum (FBS) (AusGeneX, Brisbane, QLD, Australia), and incubated in CO_2_ incubator at 37 °C.

The CVB5 strain (GeneBank: Mh201081.1) was isolated in the Kunming, Yunnan province in 2014, and preserved in our laboratory. CVB5 virus titer detection was performed in Vero cells. To investigate the effects of circRNAs on CVB5, RD (7 × 10^5^/well) was placed in a 6 well culture and infected with 1 multiplicity of infection (MOI) of CVB5 diluted in DMEM. After 2 h, the DMEM containing viral particles was exchanged with a 2% FBS medium for an additional 24 h. Control cells were treated equally but with the absence of the virus.

### 4.2. RNA Extraction, Sequencing, and the Identification of circRNAs

Total RNAs were isolated from infected, as well as uninfected RD cells, using RNAiso Plus (Takara, Tokyo, Japan), in accordance with the manufacturer’s instructions. The purity, concentration, and integrity of extracted total RNAs were evaluated using a NanoPhotomete (IMPLEN, Westlake Village, CA, USA), Qubit^®^ 2.0 (Life Technologies, Carlsbad, CA, USA) and Agilent 2100 (Agilent Technologies, Santa Clara, CA, USA). A total amount of 3 µg of RNA per sample was used as input material for the RNA sample preparations. Sequencing libraries were generated using a NEBNext^®^ UltraTM RNA Library Prep Kit for Illumina^®^ (NEB, Ipswich, MA, USA) following manufacturer’s recommendations. In brief, mRNA was purified from total RNA using poly-T oligo-attached magnetic beads. First strand cDNA was then synthesized using random hexamer primers and M-MuLV Reverse Transcriptase (RNase H). Second strand cDNA synthesis was subsequently performed using DNA Polymerase I and RNase H. Remaining overhangs were converted into blunt ends via exonuclease/polymerase activities. Following the adenylation of 3′ ends of DNA fragments, NEBNext Adaptor with a hairpin loop structure were ligated to prepare for hybridization. To select cDNA fragments that were preferentially 150–200 bp in length, the library fragments were purified with a AMPure XP system (Beckman Coulter, Beverly, MA, USA). Then 3 µL of USER Enzyme (NEB, Ipswich, MA, USA) was incubated with size-selected and adaptor-ligated cDNA at 37 °C for 15 min, followed by 5 min at 95 °C before PCR. PCR products were purified (AMPure XP system) and library quality was assessed using an Agilent Bioanalyzer 2100 system.

The clustering of index-coded samples was performed on a cBot Cluster Generation System using a TruSeq PE Cluster Kit v3-cBot-HS (Illumia, San Diego, CA, USA) in accordance with the manufacturer’s instructions. Following cluster generation, the library preparations were sequenced on an Illumina Hiseq platform and 150 bp paired-end reads were generated.

Human genome sequences and gene annotations were downloaded from the NCBI database (https://www.ncbi.nlm.nih.gov/projects/genome/guide/human/index.shtml accessed on 12 June 2019). The paired-end clean reads were aligned to the reference genome using find_circ and CIRI2, and then junction reads at the same location were then identified as circRNA [28].

### 4.3. Enrichment Analysis of Differentially Expressed circRNAs

Expressed circRNAs were quantified by TPM (transcript-per-million). Differential expression analysis of two groups (three biological replicates per condition) was performed using the DESeq2 R package (1.10.1) [29]. CircRNAs with *p*-values < 0.05 and fold changes (FC) > 2 were defined as differentially expressed

Gene Ontology (GO) enrichment analysis of differentially expressed circRNAs and statistical enrichment in KEGG pathways were implemented by the cluster Profiler R package 2110. CircRNAs with *p*-values < 0.05 were considered to be significantly enriched [30].

### 4.4. Real-time Quantitative PCR

Total RNAs were extracted with RNAiso, plus (Takara, Tokyo, Japan) mock-infected and CVB5-infected RD cells. Reverse transcription was carried out using Hifair^®^ Ⅱ 1st Strand cDNA Synthesis SuperMix for qPCR (Yeasen Biotechnology Co., Ltd., Shanghai, China) in accordance with the manufacturer’s protocol. Thereafter, qPCR was performed using Hieff^®^ qPCR SYBR Green Master Mix (Yeasen Biotechnology Co., Ltd., Shanghai, China) on a 7500 real-time PCR system programmed with the following cycling conditions: 95 °C for 5 min, followed by 40 cycles of 95 °C for 5 s, extension and annealing at 60 °C for 34 s. The 2^-ΔΔCt^ method was used for quantification. The expression levels of circRNAs were normalized to those of ACTB. Primer sequences were given in Appendix A. The expression levels of miRNAs were normalized to those of U6. Primer sequences were given in Appendix A.

### 4.5. Construction of Interaction Network

Putative miRNA binding sites in six circRNAs that were enriched in the innate immune pathway, based on GO and KEGG analyses, were identified by employing the miRanda algorithm. CircRNA-miRNA-mRNA interaction networks were constructed using Cytoscape (v3.4.0) [31].

### 4.6. CircRNA Plasmid Construction and Transfection 

The cDNA of circRNA0001066 and circRNA0002006 were cloned into the pcDNA3.1 vector to yield pcDNA3.1_circRNA0001066 and pcDNA3.1_ circRNA0002006. Transfection was carried out using the TransIntroTM EL Transfection Reagent (TRANSgene, Beijing, China) in accordance with the manufacturer’s instructions.

### 4.7. Western Blotting Analysis

Total protein was extracted by radioimmunoprecipitation assay (RIPA) lysis buffer (Solarbio, Beijing, China), and the protein concentration for each sample was calculated by a bicinchoninic acid (BCA) protein assay kit. Sodium dodecyl sulfate polyacrylamide gel electrophoresis (SDS-PAGE) for protein separation and polyvinylidene difluoride filter (PVDF) membranes were used to transfer separated proteins. PVDF membranes with protein bands were blocked with 5% (*w*/*v*) nonfat dry milk, and probed with anti-α-Tubulin antibody and anti-CVB5 VP1 antibody. Then, the membranes were incubated with diluted secondary antibodies for 1 h at room temperature. Eventually, the membranes were scanned by chemiluminescence image analysis system Tanon 5200 (Tanon, Shanghai, China).

### 4.8. Statistical Analyses

Data are presented as means ± SD. CircRNA expressions were compared between CVB5 infected-cells and controls using Student’s *t*-test. The expressions of mRNA and miRNA were compared between control cells and circRNA overexpression cells using Student’s *t*-test as well. *p*-values < 0.05 were considered to be statistically significant.

## 5. Conclusions

In this study, we identified, for the first time, the expression profiles of circRNAs that are associated the interaction of CVB5 and host cells, and constructed circRNA-associated regulatory networks relating to a host’s first line of defense against viral infection—innate immunity. We further investigated the role of circRNA on the replication of CVB5 and the possible mechanisms underlying its antiviral effect. These findings provided new insights for our understanding of virus–host interactions, and revealed that circRNAs may play an important role in CVB5 infection.

## Figures and Tables

**Figure 1 ijms-23-04628-f001:**
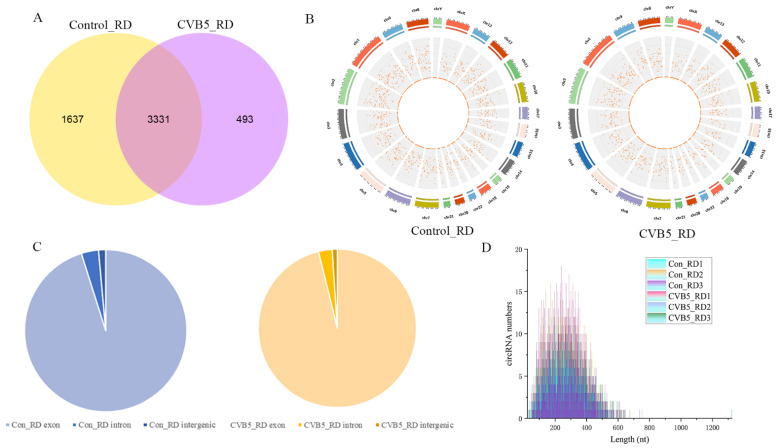
Overview of circular RNAs (circRNA) expression. (**A**) Venn diagrams presenting the circRNAs found in CVB5 mock-infected and CVB5-infected RD cells. The analysis comprises 4968 and 3824 circRNAs, identified in mock-infected and infected RD cells, respectively. Of these, 3331 circRNAs overlap between the two groups. (**B**) The distribution of circRNAs in different chromosomes. (**C**) The proportion of different categories of circRNAs. Most of the circRNAs were derived from exon regions of their host genes, although a few circRNAs were formed by circularization of intronic or intergenic sequences. (**D**) Distribution of circRNAs lengths.

**Figure 2 ijms-23-04628-f002:**
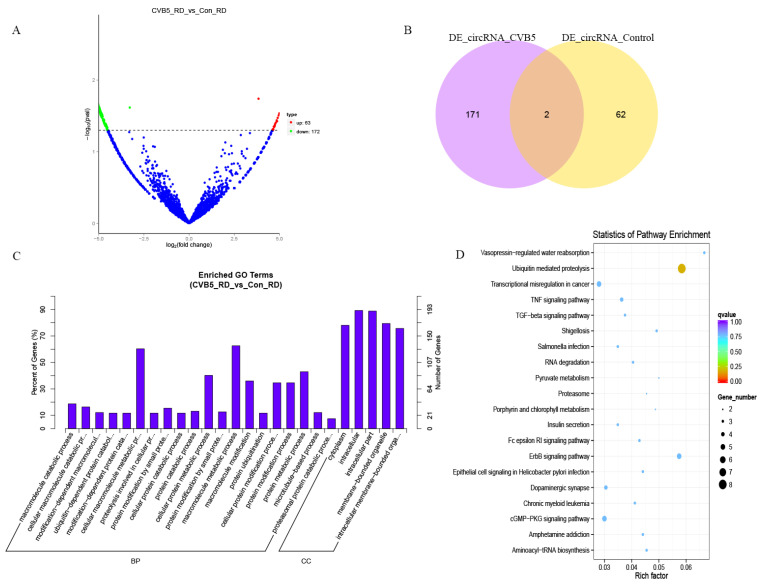
Functional enrichment analysis of differentially expressed circular RNAs (circRNA). (**A**) Volcano plot upon CVB5 infection. Red dots represent upregulated circRNAs and green dots represent downregulated circRNAs. (**B**) Venn diagram showing a comparison of differentially expressed circRNAs. The analysis comprises 173 upregulated and 64 downregulated circRNAs, respectively. (**C**) KEGG enrichment analysis, the x-axis shows the GO term, and the y-axis shows the percentage of genes and gene numbers. (**D**) GO functional enrichment analysis. The x-axis shows the enrichment factor, and the y-axis shows the pathway names. The point size represents the number of genes enriched in a particular pathway.

**Figure 3 ijms-23-04628-f003:**
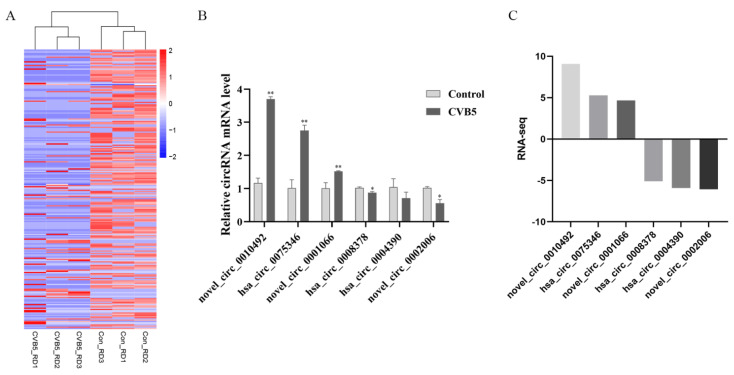
Validation of differentially expressed circular RNAs (circRNA). (**A**) Heatmap and clustering analysis of differentially expressed circRNAs. Each row represents one circRNA and each column represents one sample; −2, −1, 0, 1, and 2 represent fold change. Red indicates high expression and blue represents low expression. Con_RD1-Con_RD3 represent three mock-infected samples; CVB5_RD1-CVB5_RD3 represent three CVB5-infected samples. (**B**) Expression levels for six candidate circRNAs by qRT-PCR. Data are representative of three independent experiments with similar results. Data are shown as mean ± SD, *n* = 3 biologically independent experiments (**B**). * *p* < 0.05 and ** *p* < 0.01, two-tailed unpaired Student’s *t* test. (**C**) Expression levels of RNA-seq for six candidate circRNAs.

**Figure 4 ijms-23-04628-f004:**
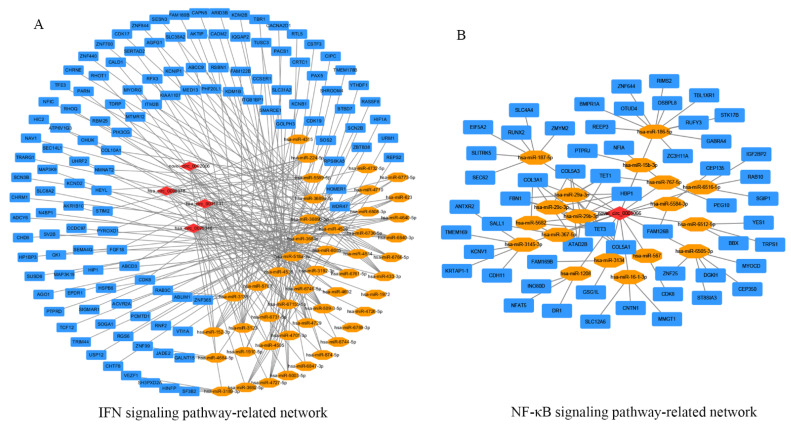
Construction of circular RNAs (circRNA)-microRNAs (miRNAs)-mRNA interaction networks. (**A**) Network showing the IFN signaling pathway; (**B**) network showing the NF-κB signaling pathway. Rhombi represent circRNA, ellipses represent miRNA, and rectangles represent mRNA, respectively.

**Figure 5 ijms-23-04628-f005:**
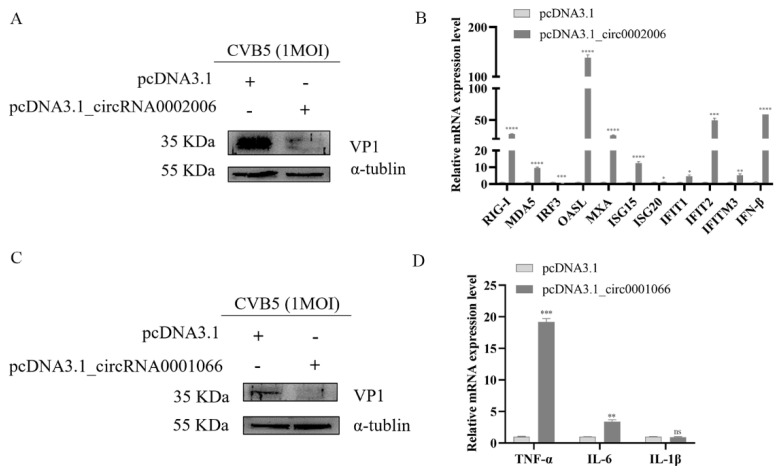
Exploring the effects of candidate circular RNAs (circRNA) on CVB5 infections and mRNAs associated with enriched pathways. (**A**) Western blot assay of CVB5 VP1 expression in RD cells overexpressing novel_circ_0002006; (**B**) RT-qPCR detection of the expression of mRNAs related to IFN signaling pathway; (**C**) Western blot assay of CVB5 VP1 expression in RD cells overexpressing novel_circ_0001066; (**D**) RT-qPCR detection of the expression of mRNAs associated with the NF-κB signaling pathway. Data are representative of three independent experiments with similar results (**B**,**D**) and shown as mean ± SD, *n* = 3 biologically independent experiments (**B**,**D**). Ns, not significant, * *p* < 0.05, ** *p* < 0.01, *** *p* < 0.001 and **** *p* < 0.0001, two-tailed unpaired Student’s *t* test.

**Figure 6 ijms-23-04628-f006:**
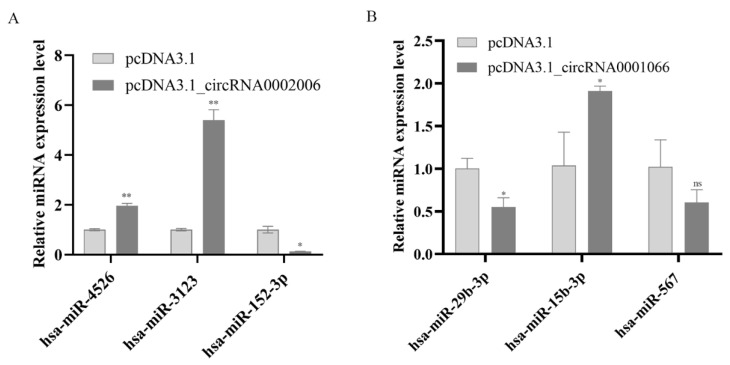
Investigating the effects of candidate circular RNAs (circRNA) on human microRNAs (miRNAs) expression in circRNA-miRNA-mRNA interaction networks. (**A**) RT-qPCR effects of novel_circ_0002006 on human miRNA expression; (**B**) RT-qPCR detection of the effects t of novel_circ_0001066 on human miRNA expression. All Data are shown as mean ± SD, *n* = 3 independent experiments with similar results. Ns, not significant, * *p* < 0.05 and ** *p* < 0.01, two-tailed unpaired Student’s *t* test.

## Data Availability

The genetic data presented in this paper are publicly available in the GenBank database under accession No. GEO180816.

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
