# Peer review of "Bioinformatics and Screening of a Circular RNA-microRNA-mRNA Regulatory Network Induced by Coxsackievirus Group B5 in Human Rhabdomyosarcoma Cells"

_ijms, 2022, doi:10.3390/ijms23094628_

Round 1

Reviewer 1 Report

In general, This paper is a good paper to publish since circRNA is a new RNA type that needs to be discovered. It is a combination of bioinformatics and wet lab paper.

Scientific suggestions

  • I wonder is there a way to differentiate host circular RNA vs virus circRNA? If yes, can you please separate them in a table and show it as a supplementary figure? Because in line 184 you mention something similar concept to what I asked.
  • Since this paper focuses more on circ0002006 and circ0001066 can you give the list of miRNA and mRNA targets as a supplementary table?
  • Can you show your finding for differentially circRNA has how much part of exon-intron or intergenic sequences? That information can be beneficial for future research.
  • Since this is a new RNA type I hope to see a figure which depicts the circRNA of both host and virus and shows inhibition of miRNA and mRNA-protein (analogy).
  • In discussion, you mention CVB5 replication when you talk about WB results in figure-5, can you say CVb5 infection instead of replication that word is misleading.

In the methods section,

  • You say MOI is 1 but in figure 5a and 5C, you say 0.1 which one is correct?
  • You stated that KEGG pathway p values p<0.05 but line 188 states 0.5 please correct that.
  • Some companies’ country is missing in the methods section, please correct that.
  • In your RT PCR cycle description extension step temperature and time duration of that step is missing

Format and Language

  • There are some typos here and there.
  • For example line 98
  • Line 173
  • Line 267
  • Line 156 you can use upregulate word instead of enhanced.
  • Please use either CVB5 or CV-B5 to have consistency.
  • After line 99 and line 134, there is a huge empty space.

Statistics

There is no statistical analysis done in any of the graphs which have standard deviation like Figure 3-Figure 6A and 6B, Figure 5B, and 5D.

Author Response

Thank you for all your comments concerning our manuscript,we have uploaded as an attachment.

Reviewer 2 Report

Comments for “Bioinformatics and Screening of a Circular RNA-microRNA-2 mRNA Regulatory Network induced by Coxsackieviruses 3 Group B5 in human Rhabdomyosarcoma cells”

MicroRNA and viral RNA infection is important and novel.  HFMD is important human infection and this disease are detecting in worldwide.  Obtained results are new and provide very useful information.

The MS needs revisions, see comments below.

Remarks:

Line 1-4 

Capital or not for first letter for words (human, induced, cells) 

Please use Coxsackievirus Group B5 instead Coxsackieviruses Group B5 for all text

Line 13  “infection”  please delete

Line 13-14 - the sentence starts “However …” must be deleted or revised.

Line 18 – human rhabdomyosarcoma cells

Line 18-19 - Phrase “(HFMD virus ... )” please delete or using name of strain (s) or isolate (s)

Line 22 – “235” - this number are not presented in Results (only in Discussion). Please clarified this situation.

Line 27 and below (all text and figures) Please check “novel_circ_0001006” or “novel_circ_0001066”.  It is unclear.  

Line 31 -  Hand, …

Line 48 and below:  Please revise “after-effects or aftereffects” and “CV-B5 or CVB5”

Line 142 – INFβ or INF- β

Line 147-152 - duplication of legend for Figure 5, must be deleted

Line 164 - Please correct,  “over-expression” or  “overexpression”

Line 175 – 180 - Provide a more information for circRNAs for  other viral infections  for in vitro similar models (in vivo models and in human if possible).   For example, see and use also doi: 10.3389/fcimb.2021.779919  for discussion. 

Line 178 - What type of viral infections,  RNA-or DNA infections?

Line 193 –194 - “235”, reference for Table S4, please add in this sentence

Line 200 - SH-SY5Y cell, please revise

Line 216-217 - abbreviations:  HBV, IAV JEV – may be unclear for readers

Line 262 - were HEK-293T cells used? Please clarify

Line 261 - Provide a more information for   using  cell lines (collections, collection number,  and  other. 

Line 265 - Provide a more information for   using viral strain (s) - name, collections, collection number, place and date of isolation,  GenBank accession number for viral genome  and  so on.  

Line 319-321 - Please move Tables 1-2 to supplemental

Author Response

Thank you for all your comments concerning our manuscript. We have uploaded as an attachment.

Round 2

Reviewer 1 Report

I see all the points highlighted are corrected. 

Author Response

Thank you for your comments.